# Acceptability and feasibility of video-based health education for maternal and infant health in Dirashe District, South Ethiopia: A qualitative study

**Wanzahun Godana Boynito**[1,2]*, **Godana Yaya Tessema**[3], **Kidus Temesgen**[1], **Stefaan De Henauw**[2], **Souheila Abbeddou**[2]

1 School of Public Health, College of Medicine and Health Sciences, Arba Minch University, Arba Minch, Ethiopia, 2 Department of Public Health and Primary Care, Faculty of Medicine and Health Sciences, Ghent University, Ghent, Belgium, 3 Department of Midwifery, College of Medicine and Health Sciences, Arba Minch University, Arba Minch, Ethiopia

* wanzanati2011@gmail.com, wanzahungodana.boynito@ugent.be

**Data Availability Statement:** Data used in this analysis are available at https://data.qdr.syr.edu/.

## Abstract

Evidence about innovative methods to facilitate nutrition education counseling and promote the intended behavior change at scale is limited. We assessed the acceptability and feasibility of a video-based health education intervention aiming to promote community care for pregnant women, mothers, and infants in the Dirashe District, Ethiopia. Using a phenomenological study design, the experiences of study participants in a trial testing the effectiveness of video-based health education on birth outcomes and nutritional status of mothers and their infants six months postpartum were assessed. Focus group discussions (FGDs) and key informant interviews (KIIs) were used to collect the data. The study was conducted in the Dirashe District, South Ethiopia. Five FGDs and 41 KII were conducted among video implementers, mothers, nurses, and health extension workers (HEWs) in eight intervention villages. All data were collected with a tape recorder. The tape-recorded data were transcribed and then translated into English. Data were analyzed using thematic content analysis. The videos delivered messages about nine themes on health, nutrition, and hygiene related to mothers and infants. Overall, the video-based health education interventions was acceptable and feasible. Messages delivered were found to be clear, easily understandable, culturally acceptable, and relevant to the needs of the mothers. Feasibility was affected by the nature of the work, lack of help, and overlapping duties of the HEWs. The video-based health education intervention was acceptable and feasible. It was suggested that determining a common location/venue to show the videos, involving husbands, and involving HEWs could improve the intervention.

**Trial registration:** The effectiveness "parent" study was registered as a clinical trial with the U.S. National Institute of Health (www.ClinicalTrials.gov; NCT04414527). The qualitative study included recipients from the same cohort (participating mothers from the intervention group), in addition to video implementers, health extension workers the Health Development Army, and nurses from the intervention communities.

**Funding:** This work was supported by the Flemish Interuniversity Council (Belgium, VLIR-UOS), which funded the research through its Institutional University Cooperation (IUC) coordinated by KU Leuven (Belgium) and Arba Minch University (Ethiopia) (https://www.vliruos.be/en/projects/project/22?pid=3604). This qualitative research was funded through the Global Minds Fund of Ghent University https://www.ugent.be/en/research/funding/devcoop/globalmindsfund.htm (GRANT GMF.CAB.2021.0030.01). The funders had no role in the study design, data collection and analysis, decision to publish, or the preparation of the manuscript.

**Competing interests:** The authors have declared that no competing interests exist.

## Introduction

The United Nations (UN) articulated the determination and commitment of the member states to end preventable child and maternal deaths by 2030, as specified in Sustainable Development Goal 3 (SDG 3) [1]. Antenatal care (ANC) is a critical strategy for reducing maternal mortality, as it facilitates the identification and mitigation of risk factors early in pregnancy and postpartum [2]. Appropriate, timely, and frequent use of ANC enables the delivery of essential services, including malaria treatment, immunization, nutrition and health counseling, and micronutrient supplementations [2–6]. Health service delivery through the health extension program makes care for mothers and children accessible to the community. The home-to-home visit strategy enables the identification of early pregnancies, and the ONE to FIVE network, a group of six people, with one person acting as the leader who coordinates the other five people in a specific activity, supports nutrition and health educational interventions that promote preventive health behaviors to improve maternal and neonatal health through better knowledge, attitudes, and practices [7].

Ethiopia adopted and implemented the ANC guidelines of the World Health Organization (WHO) in 2004 [3]. The guidelines recommend at least four antenatal visits for women with low-risk pregnancies, and evidence-based content, including iron and folic acid supplementation, must be provided for each visit. The maternal mortality rate in Ethiopia remains high (420 maternal deaths per 100,000 live births in 2016) [7, 8]. The national Demographic and Health Survey (DHS) indicated that 62% of mothers attempted at least one ANC visit during pregnancy in 2016, only 66% of mothers received nutrition counseling during the ANC visit, and 70% reported difficulties accessing health services. This is indicative of the disparity that exists between what is required by a growing population and a society's service capacity. Health services must be made accessible to the community, and different approaches should be adopted [7–9]. The limited coverage of ANC is caused by the lack of logistical resources, of training provided to service providers, and of clear policy directions in the service delivery system [9]. In many low- and middle-income countries, including Sub-Saharan Africa, health care providers do not routinely provide women with information as part of ANC or fail to provide information in a way that is understood by and is practical for women [7, 10–12]

Acceptability, fidelity and feasibility influence sustainability and scalability of any intervention. Hence, these factors should be considered at different stages of the intervention development, evaluation and implementation, in different settings and over time [13–15].

Success of behavior change communication (BCC) in improving maternal and child nutritional status and in increasing adherence to recommended practices during pregnancy, lactation and early childhood are largely determined by the importance given by the community and the health workers [16]. Several platforms to deliver BCC programs are currently tested. Using cordless projectors and locally created videos gives organizations more quality control over the end message, expand the number of people reached, allow for the use of non-expert facilitators, and allow for contextually appropriate information [17]. They can also be used in areas without access to electricity, helping to bridge the digital divide, and serving as a leapfrog technology for areas that would otherwise not have access to media. Traditional methods of dissemination such as pamphlets and flipcharts exclude people with low literacy and are not environmentally conscious. Fatigue over repeated messages is also a concern in behavior change interventions. In addition, messages of video-based education improve retention of the intended behavior, its acceptance and adherence [17–19]. Our knowledge about innovative methods that can facilitate NEC and promote the intended behavior changes at scale, and their acceptability and feasibility, is limited. A cluster randomized controlled intervention study was conducted in Dirashe district Southern Ethiopia with the aim to assess the effects of video-

based health education on nutritional status of pregnant mothers and their infants (from 0 to 6 months). To provide insight into the acceptability and feasibility of video-based BCC, we investigated the beliefs, attitudes, needs, and situations of video implementers, beneficiaries, and health extension workers (HEWs) at the end of the study. This qualitative study aimed to assess the feasibility and acceptability of video-based health education in a local context and to identify the opportunities and challenges of its implementation.

## Methods and materials

### Study setting

The study was conducted in the Dirashe District, Segen Area People's Zone, Southern Nations, Nationalities, and People's Region (SNNPR). This study engaged in a qualitative evaluation of a cluster randomized controlled trial (cRCT), which was implemented in the district. The cRCT included two cohorts of 580 eligible pregnant women who were followed from three months of pregnancy to six months postpartum. Recruitment of the study participants started October 2020 and continued till January 2021. However, the follow up period was till February 2022. The qualitative research was conducted at the end of the parent study during February 2022. Rural communities were cluster randomized at the Health Extension Worker's (HEW) level to receive health and nutritional education in a video-based approach (Health-Video) or in a standard form (Control, standard counseling). Women in their first pregnancy trimester who reside in rural kebeles of Dirashe District and with a plan to stay for the next 12 months (six-month pregnancy and six months postpartum), were invited to the health center and were recruited if 1) they signed an informed consent form, 2) were at least 18 years old, 3) were permanent resident of the village of the study intervention/control, 4) planned availability during the whole period of the study (12 months); 5) accepted the intervention package including home visits for data collection and morbidity follow up.

The study did not include women with severe anemia (hemoglobin <70 g/L), under nutrition (defined as body mass index before pregnancy of <18.5 kg/m$^2$), chronically ill mothers with tuberculosis or other chronic diseases, and reportedly HIV-positive. In addition, individuals with anatomical deformity were excluded due to the difficulty of measurement of height.

The intervention cohort (Health-Video) received innovative video-based nutritional and hygienic education, and the second cohort (Control) received national standard counseling. Intervention videos were produced by a local team after intensive training with illuminAid (formally One Mobile Projector per Trainer, OMPT) (https://www.illuminaid.org/). The final version of the videos was reviewed by the study investigators, project members, and the illuminAid. The videos were projected every two weeks at the home of the participating women, and monthly in forum or in group, at nearby health facility, church or at the school. The ten videos implemented covered nine themes about the benefits of taking iron and folic acid tablets (IFA) during pregnancy, vaginal hygiene and perineal care, practices to avoid important worm infections, hygienic practices when preparing food for the mother and her family, additional food intake and diversification, health service utilization (including the importance of deliveries at the hospital with a maternity waiting area [MWA]), colostrum feeding and early initiation, breastfeeding practices and EBF, maternal safety during pregnancy and breastfeeding, and safety precaution during pregnancy. In addition, both groups received national nutrition and health care information, including IFA supplementation, treatment of any symptomatic illness, and deworming in case of symptomatic complaints during the second and third trimesters. Follow up visits of pregnant women at the health centers were done at 6 and 9 months pregnancy, and for the pair mother-infant were carried out within one month after delivery and monthly until 6 months postpartum.

## Study design

A phenomenological qualitative study approach was carried out to explore the acceptability and feasibility of video-based health education among pregnant and lactating women and its barriers and facilitators.

## Study participants

The study population included mothers, video implementers, HEWs, the Health Development Army (HDA), and nurses from the intervention communities.

## Sample size and sampling

Three focus group discussions (FGDs), including 10–12 participants per group with mothers with good adherence to video attendance and two with mothers with poor adherence were conducted until data saturation. Furthermore, eight video implementers, eight HEWs, five HDA mothers, and 16 mothers (two per intervention kebele) were involved in key informant interviews (KIIs). In addition, four nurses were interviewed. All participants were selected using a purposive sampling technique.

## Data collection tools

A pretested semi-structured interview and a FGD guide were used to collect the data. After pretest, unclear questions were corrected or modified. Checklists were used before conducting the discussions and interviews.

## Data collection

Data were collected at the end of the parent study during February 2022. Tape-recorded semi-structured interviews were conducted following both FGDs and KIIs guided by an interview guideline and supplemented by follow-up and probing questions. FGDs and KIIs were conducted after the participants provided written informed consent. The FGDs were moderated by experienced facilitators and assisted by an experienced person from the same culture, to facilitate the quality of data collection in a local language. Field notes were also used to amend the audio-recorded data. All field data were collected in the nearby health post where it is possible to conduct the discussion in optimal conditions. Each FGD lasted from 1hour 25minutes to 1hour 45minutes, while the KII lasted between 35 min—55 min. Data collection was terminated after saturation.

## Data quality assurance

To ensure the quality of the KII and FGD data, different mechanisms were used, including recruiting data collectors and facilitators who had experience conducting qualitative studies. Two days of training on how to conduct KIIs and facilitate FGDs overseen by senior qualitative research experts were given before fieldwork. A senior public health expert supervised the overall process daily during the fieldwork. The quality control of the transcripts and translation was done on 10% of the audio records by the same senior qualitative research expert.

## Operational definitions and definitions of terms

***Semi-quantitative reporting:*** "Majority" is used to indicate more than 75% repetition of the theme, "many" is used for 50–75%, "some" or "several" represents 25–49%, and "a few" represents less than 25% [18].

*Acceptability*: This was measured using conventional methods, such as adherence, cultural support, clarity, effectiveness, and applicability.

*Feasibility*: This was measured by the cost required to participate in the program, the time to participate, and overlapping duties that may hinder participation.

*ONE to FIVE Networks*: These are a group of six people, with one person who has better leadership skills acting as the leader who coordinates the other five people in a specific activity.

*The Health Development Army (HDA)*: The HDA is made up of volunteers who are the leaders of the ONE to FIVE networks [19].

*Health extension workers (HEWs)*: These are female community health workers working at the kebele (village) level in the Ethiopian health system.

**Good adherer:** Participating mother who attended at least 75% of the video projections as per their schedule.

**Poor adherer:** Participating mother who attended less than 75% of the video projections as per their schedule.

## Data analysis

All KIIs and FGDs were captured using voice recorders, and field notes were transcribed verbatim into local language and then translated to English by KII/FGD field facilitators every day. The translation of transcripts was done by experienced data collectors and supervisors and back translation was done to ensure the validity of the translations of the transcripts. The transcripts were checked independently by the supervisors for verification. The data were analyzed through thematic content analysis. Major themes were derived based on the study's objectives. However, sub-themes were derived from the text itself through repeated reading by the research team. After reading the transcripts, the emergent themes were identified and then coded for each theme to specify individual topics identified during the discussions. Transcripts were coded by two research team members (WG and GY) using Quirkos qualitative software, version 2.1.

Statements were grouped by codes according to corresponding themes. Once themes were established, the transcripts were reread to ensure that the themes appropriately reflected the content of the data. All identified themes were confirmed by the researchers to capture discussions from the KII/FGDs. The findings were presented in narratives by thematic areas, based on the objective of the study. The quotes included in the results are typical opinions and views expressed in each KII/FGD to exemplify emergent themes.

## Ethics

Ethical approval for the study including the qualitative research, was obtained from Arba Minch University, College of Medicine and Health Sciences Institutional Research Ethics Review Board (IRB/158/12 dated January 17, 2020) and Ghent University Hospital (UZ Gent BC-06756). The parent trial was registered in the clinical trial registry (NCT04414527). The qualitative study included recipients from the same cohort (participating mothers from the intervention group), in addition to video implementers, health extension workers the Health Development Army, and nurses from the intervention communities. FGDs and KIIs were conducted after all the participants provided written informed consent. Privacy and anonymity of the study participants were respected.

## Results

### Sociodemographic characteristics of the study participants

All study participants were women aged between 22 and 50 years from the eight intervention kebeles. They were involved in a parent trial that tested the effectiveness of video-based health

**Table 1. Sample size and summarized methods used in the qualitative study.**

| Data collection method | Target group | Sample size | Remark |
|---|---|---|---|
| Key informant interview (KII) | Mothers with good adherence | 8 | One mother per kebele |
| | Mothers with low adherence | 8 | One mother per kebele |
| | Health Development Army (HDA) | 5* | One HDA per kebele who has experience |
| | Video implementers | 8 | One per kebele |
| | Health extension workers* | 8 | HEWs per kebele |
| | Nurses | 4 | Site supporting nurses to the video implementers at the kebele |
| Focus Group Discussions | Mothers with good adherence | 3 | One FGDs per group |
| | Mothers with low adherence | 2 | One FGDs per group |

*Data saturation is one of the criteria to stop the interview

education on birth outcomes and the nutritional status of mothers and their infants at six months postpartum. The participants included trial participants, HEWs, HDAs, video implementers, and nurses (**Table 1**).

## Acceptability of video-based health education among pregnant and lactating women

The acceptability differed according to the subject covered in the videos and the support provided by the video implementers. The majority of the interviewees reported that the videos were clearly presented, in the local language, culturally adapted, and understandable.

> *"There is something I like. Thank you for teaching us from the very beginning of our pregnancy and for giving us the information on immunizations and follow-up needed for our children. I used to give birth in a health care facility. What they showed us on TV* [projector] *taught us how to take care of a pregnant mother. They also taught us how to wash hands and breastfeed babies. The video message was clear and easily understandable. The language in which the video was made is our own."* (GA0201)

Moreover, the participants reflected on the relevance of the videos and their applicability in their local and personal contexts.

> *"The videos were in our own language, and they are easy to understand and practical. I would love it if we all apply* [the messages] *to our day-to-day life, which is not difficult in practice."* (FGD 0801)

The use of video in antenatal and prenatal care was perceived as an opportunity and as an acceptable means of behavior change communication at the community.

> *"No such video has been shown in the neighborhood before. I'm glad this is the first time. They used to come and teach the mothers. As I worked as a traditional birth attendant before it was banned, the mother had no such opportunity to see health information using drama or roleplays. They are very lucky. They assigned two girls* [video implementer and household data collector] *to follow up with the women without HEWs. The girls came to me to tell me to advise the mothers who were not going to watch the videos. I am commonly using phrases like 'consider this as lottery' since this opportunity was not given to all mothers."* (IDI 0403)

Most of the time, the videos were displayed at home by the mothers. However, in some cases, the videos were also displayed in neighbors' homes and in common areas, such as churches, health posts, and maternity waiting areas of nearby health centers.

*"The biggest problem is showing* [videos] *home to home, and some of the houses are not appropriate. Sometimes, we appoint the mothers to show the video at the school. But schools do not allow* [this activity], *and even in some cases, it is too noisy to show the videos. We also tried churches, but it was still not good. So, it's important to consider where to show the videos."* (IDI 0601)

The extent to which the learned behavior is linked to the health of the beneficiaries may affect acceptability. However, in this study, mothers unanimously liked the videos, as they are linked to their health, but showed also some preferences.

*"The women said they liked most of the videos but loved the video of giving birth in a health facility. Because, they say that they can be taken care of in an ambulance immediately in case of emergency and this is a lifesaving."* (IDI 0301)

## Implementer's related factors in video-based health education among pregnant and lactating women

The mothers perceived that the projections made by the video implementers were not always of good quality, and they reported that the low technical skills of the video implementers might have been the cause. They also complained about the limited availability of video implementers.

*"As I told you on different occasions, there are many mothers who have said why we are not allowed to watch the videos. I was also asking if the video girls could show the videos to all pregnant women, but they said they had been given a list of mothers to whom they should show the videos. I don't think it is good to treat women in the same community differently. It is better to think about it."* (IDI 0403)

Provider's support enhances acceptability and adherence to a recommendation. Empathy of implementers was captured by their willingness to show the video to a participating mother who missed a visit, and their behavior towards her during the following video show session. The shortage of time for the video implementers was among the factors that the mothers did not appreciate and recommended improvement.

*"I also feel that it would be better if all mothers could see the videos, and they said* [video implementers] *that they have no time to show all the mothers in the community. For me, I am very lucky to be part of it."* (FGD 0302)

## Feasibility of video-based health education among pregnant and lactating women

The feasibility of the video intervention was assessed using proxies of 1) cost to the mothers related to the program, 2) time to take part in the video, and 3) lack of help and overlapping duties. In most of the sessions, the recurrent challenge was a shortage of time to take part in the video program.

Affordability or financial acceptability is one of the key indicators for utilization of a given service. Beneficiaries were asked if financial access affected the overall acceptability. Indirect costs like time compensations were an indicator to assess cost of a program.

*"No payment is needed. But one day, the video girl called us to come to the health post to show us the videos, but some mothers did not come on time, and it was working time* [for me], *and I was very angry at her."* (FGD 0303)

*"Due to a shortage of time, it would be best if they can show us once per month by combining the videos, and if the day is Sunday afternoon, when we have rest. The problem is that the video girls come the day they want, and sometimes they do not consider our work."* Onota (Good adherer)

*"Some of our mothers complain about not having time to watch the videos. We are convincing them, but there are still problems. The major problem is with the mothers, who do not have someone helping them."* (IDI 0703)

Lack of support for household and farm activities is identified as a hindering factor for adherence to the video schedule and latter its practicability. Lack of support results also in less time for the mothers to attend the video projection sessions, affecting their adherence and the feasibility of the program.

*"The only thing they find difficult to watch* [the videos] *is the pressure of work at home. As you know, sometimes when it comes to the working days, it is the mother who needs to stay at home and do all the house activities, as well as prepare food for the day and the next day. Most of the time, the complaint is from a workload."* (IDI 0101)

*"There is nothing wrong with practicing the lessons from the video. But, since we don't have a person who helps my husband, I am forced to work and cannot attend many sessions."* (GA 0202)

All the study participants discussed and mentioned that there was no cost to participate in the video sessions.

*"We did not pay, and it is free. But sometimes, when you participate in the forum, it can cost you since you will spend the whole day there in the meeting."* (FGD 0405)

*"They* [mothers] *did not request a motivation fee for watching the video. Whatever was provided was for their benefit, and the payment should not be an issue."* (IDI 0201)

## Facilitators of video-based health education among pregnant and lactating women

The facilitators of the video-based intervention were the health facility, the community, and the HDA. The community and the HDA helped the participants continue to follow the videos. The support provided by the health facility was one of the most important themes raised.

*"We were collaborating with the HEWs, and they helped us. HEWs are well respected, and they convinced the husbands."* (IDI 0101)

*"It is a good idea to have a continuous view of the video. The person who projects the video also explains the importance of each session and video. It would be better if they could bring all the mothers together so that they could learn from each other. It may also be good to consider including family and friends or other households."* (GA 0201)

## Barriers to video-based health education among pregnant and lactating women

Barriers and challenges to the video projections included access to power to charge the portable projectors, mothers' attitudes, place to display the videos, husbands' attitudes, the noise in the vicinity, and disturbances from children.

*"Sometimes we don't have electricity, and the battery of the projector is down."* (IDI 0201)

*"One day while we were watching, it went off* [battery of the project], *but it's good that the videos are repeated* [in other sessions]*."* (FGD 0408)

Mothers' attitudes toward the videos were also one of the challenges to the program.

*"Some of the mothers deliberately miss the video projection day. The attitudes of the household are also a challenge."* (IDI 0201)

The type and size of some houses, the continuous disturbances from the children, and the low attention or focus on the video(s) made it difficult to display the video.

*"Children in the community also disturbed when the video girl came to show the video."* (GA 0201)

*"Despite the importance of the videos, we have many problems, and sometimes, when you are alone at home and have lots to do and children are around, it is a challenge to follow, even if it was displayed in our language."* (FGD 0309)

The noise from cattle and some visitors are also barriers.

*"I agree with my sister's comment about disturbances. Yes, kids are disturbances, and also the cattle. Regarding the church, the church may have its own program, and it may not be good as well. So, for me, the household and church are almost similar and are not a good option."* (FGD 0312)

*"I would say that workload is a major barrier for mothers to watch the videos. The lack of a uniform display area or home, which is not similar for all households, can also be an issue. Sometimes, when it is very difficult to show the video in the mother's house, we arrange to show the video in the neighbor's house if the room is good. But this alternative has its own limitations. We cannot control the family and children in the neighbor's house where we arranged the projection, and sometimes they disturb us."* (IDI 0601)

Husbands' attitudes have also been reported to be a limitation when they are not supportive. However, some show support for their wives. Partner involvement and support differed among the participants.

*"We had a conflict with my husband the day before the video display, and when she* [the video implementer] *came to show the video, he started shouting at her and telling her that after you showed her the video(s), she started to not work and said* [his wife] *that you told her to have rest? It was not good day, and finally, the HEW told him that it is important for me, and he started to allow me."* (FGD 0301)

*"One day when we showed the video, one of the husbands started to quarrel and said, 'Did one of our fathers learn how to give birth to us?' He refused and prohibited his wife from participating. His wife did not follow the video to the end."* (IDI 0601)

On the other hand, there are husbands who support their wives to attend and follow the lessons shown in the video sessions. This will in turn affect the uptake of video messages and the achievement of the objective of the behavior change communication.

*"My husband even encouraged me. But we are in the same kebele, where some husbands are supportive and others are not. I think if there were a similar supportive attitude among all, it would be good."* (FGD 0305)

### Applications to day-to-day life

The benefits of the videos can also be judged by their application to day-to-day life and their contribution to the health of the community.

*"Yes, I think the program has made a difference. One day, I went to a mother's house, and when I asked her how she was, she replied, 'I had anemia, and my blood started to return after taking the pill.' This is the benefit of the video* [IFA] *and the practice they implemented."* (IDI 0601)

The goal of the behavior change communications was to help the community to apply lessons learned to their daily life. Knowledge of the benefits and ability to apply the learned behavior depends on how the information were presented to the community. She added that *". . . there is no difficulty in implementing the change. It's easy. But I think the problem is that illiteracy has a negative impact on women because an educated mother does what she learns. The uneducated applies very little. This is what we get when we ask for feedback."* (IDI 0201)

However, some of the mothers raised the issue that the application of the video messages is also affected by support from the families.

*"For me, everything can be practiced, but it needs the support from family and friends. For example, if we are talking about the workload, it needs family support. Not only this, but even to participate in the session, you need the time that otherwise you are using for work, and this needs your husband or family support."* (FGD 0303)

Almost all the participants agreed that the application of video messages to day-to-day life is very important. However, implementation varies among mothers.

*"There is nothing that I have not done. I went to the maternity waiting room and gave birth at the health facility. I am also taking my baby to the HEWs when he is sick, and I am washing my hands before preparing food."* (GA 0201)

*"To add to what she told you, there was a pregnancy disease called anemia, and drugs for its treatment* [IFA] *are given for free. But, what we have been informed in the community regarding this was that the drug causes nausea, discomfort, and vomiting. So we did not take the drug as planned in our previous pregnancies. But, the video explained how to reduce unnecessary side effects and benefit from the drug like taking the drug after a small meal. I*

*can confirm that the benefit is large since we can easily avoid the side effects with minimal adjustments, such as taking the drug with food and before going to bed. I had no problem and gave birth, and we are here today,"* (FGD 0803)

Acceptable behavior change has a high chance of adoptions. Benefits of the learned behavior and the ease of application enhance further applicability and adoption.

*"What mothers learn from the video can be easily applied. They say that the video is very good and easily applied to their lives. Despite the workload, most of them said that they apply what they learned."* (IDI 0401)

*"It is not difficult to apply. All the videos are in our language and culture. I think all the lessons learned from the video are very good and easy to apply. The good thing is that they made it using the cultural context, including the language, the way of living, and even the clothes. But they complain about the additional food that they do not have money to buy; what can they do? Some of the mothers also ask their husband to acquire for them what is in the videos, but they refuse by saying it leads to additional costs."* (IDI 0603)

## Acceptability of the overall program

The acceptability of the video intervention was among the commonly raised themes in the discussions and interviews.

*"I would like to thank those who prepared these videos in our own language and brought them to our home to show us. I liked the video very much. All the videos were good, especially the one regarding feeding, nutrition, and handwashing. I had given four births before. In our culture, they say that pregnant women should not eat additional food during pregnancy, the reason being that additional food will help the baby to grow and predispose the mother to a risk during birth or C-section, and mothers might even die. But when I watched the video of the mother who was saying a pregnant woman should not eat additional food, the other said she had to. The discussion even debated how many times or how much additional food, like one or two more meals. She also explained that the food would give her energy during labor."* (GA0801)

Satisfaction with the content of the video, complexity, the comfort it provides, mode of delivery, and credibility of the presenters and also the messages affect the application of the videos. It was repeatedly raised that the degree to which each video is applied depends on the type of video.

*"Handwashing especially does not cost us, so we can easily apply the messages* [the video messages]. *We have lots of water. For food, sometimes, you may not have the capacity to buy. First-time pregnant women are always advised not to eat more food because our culture says that the baby will be large, and the mother will face problems when giving birth. But now, thanks to the videos, we have been taught the correct information. I am very happy to have been part of it, and it would be good if others could also see the video and benefit from the information provided. The other thing that I found important was the first milk to the baby and only giving birth at health institutions."* (GA 0501)

Acceptability or perception among implementation stakeholders that a given video-based health education is agreeable, or satisfactory is among the key indicators for sustainability.

Furthermore, acceptability promotes the implementation and application of lessons that are learned.

*"All* [messages] *are easy to implement. The videos helped us follow antenatal care and visit health services. They were also important in getting prepared for where the baby would be born."* (FGD 0206)

Adherence to the messages was also verified through a change in practice(s). Adherence is facilitated by personal motivation and could also be socially- motivated. Socially motivated adherence is the desire to change behavior to fit into the social environment which can be taken as an example.

*"When we asked the mothers whether they were adopting the practices that were communicated in the videos, they indicated that they adopted some but not others. For example, it is easy to implement the messages provided in the video on handwashing, breastfeeding, and shoe wearing. However, additional food, workload, and showers all need a decision from husbands because they need the capacity."* (IDI 0201)

## Future improvement to video-based health education among pregnant and lactating women

Sustainability of any intervention depends on how it incorporates the future improvement areas and tackling the challenges. Participants recommend improving the intervention in the following ways: 1) assigning a place for the video projection, 2) involving the husband/partner, 3) integrating the intervention into the health system through the HEWs, and 4) including videos that target the feeding of young children. One of the major improvement plans was to assign a common place that fits the needs of all the project recipients and ensures the continuation of the intervention.

*"Regarding the things to be improved, it is basically not the video but the place where we show the videos. In a narrow and small house, it is not easy to watch it freely. When mothers watch it in groups, the place should have ventilation and adequate seats."* (IDI 0501)

Sustained behavior change depends on different factors at varying levels. Peer factors are among the facilitators for the sustained change of the learned behavior. Partner involvement was among the key recommendation by the mothers to further implement the program.

*"It is better to include husbands in the session so that they know that what we are doing is for the baby and the family. The husband must support* [the mother] *in all the activities and in attending this video education."* (FGD 0806)

*"It would be good if our husbands watched the videos with us and facilitated things for us to follow the videos correctly. If we watch together, then we do not have to justify our need for more food and rest during pregnancy. On the contrary, they can easily help us because we keep doing a lot of work, including bringing firewood."* (GA 0202)

There are also some aspects in the content of the videos that need to be changed if scaling up is intended. Foreseeing the possible obstacles before considering major expansion is very important, including modification in the design and change in implementation strategies. With this regard, modifications of some contents of the video were also recommended.

*"All videos are good, but some advice provided in some of the videos is not easy to practice, such as intimate hygiene, raw meat* [not to eat it], *or taking showers in the river* [not to practice it]. *We commonly use rivers to wash our bodies. To some extent, handwashing is also affected by the nature of the work we are involved in. Women can easily wash their hands when they are at home, but this is difficult when they are at the farm. I am not clear about the idea of some videos and their applications. I am thinking that if the husbands are involved, then it can even encourage the mother to be part of the video* [intervention]." (IDI 0402)

Involving HEWs is also one of the most important and commonly mentioned themes. HEWs are the key and building blocks of the community health system of Ethiopia. Community health services should include and involve them to further enhance the acceptability of the new health program and its continuity. This is only possible with the government support to community health program or the health extension program. Financial support can allow the continuation of the program beyond the project life, while its integration in the national health strategy allows its scalability.

*"It would be nice if the video presentation would continue in the waiting maternity home to give health information to pregnant women. If it can be integrated into our work* [HEW] *and be shown at the health facility when women come in for regular visits, they can also involve husbands or Health Development Army leaders, and they can contribute to the change in the community."* (IDI 0702)

*"It would be good to keep showing the videos. The video girls say that they may not have a budget and need to stop it. Maybe they can give the equipment* [projectors] *to the HEWs, who can show the videos to our mothers. The video can even be shown by HDA since it is not complicated. It is also best to involve their husbands in video presentations."* (IDI 0303)

## Discussion

In the context of the cRCT "Effects of video-based health education on the nutritional status of pregnant mothers and their infants (from 0 to 6 months) in Dirashe District, Southern Ethiopia", ten videos covering nine themes around maternal and child nutrition, health and hygiene were projected during the 12 months intervention. The current qualitative research investigates the beliefs, attitudes, needs and situation of the video implementers, the mothers, and the health officials.

### Acceptability and its indicators

Video-based health education was accepted by the participants and the health stakeholders at the health service and community levels. The video-based intervention was reportedly feasible for implementation in the community through the existing community health system.

It has been hypothesized that messages related to health, food, agriculture, or infectious diseases provided in the form of videos are better understood, retained, and implemented, which also results in higher adherence to recommended practices [17–19]. This result is reached if the intervention is accepted.

The success of BCC in improving maternal and child nutritional status and in increasing adherence to recommended feeding practices during pregnancy, lactation, and early childhood are largely determined by their importance, as attributed by the community and health workers [19]. Furthermore, the success of these practices lies in their uptake by the participants and their adherence [20–22]. Several platforms for delivering BCC programs have been tested for

acceptability and feasibility. A feasibility study using a community-led video approach to promote maternal, infant, and young child nutrition in Odisha, India, explored the retention and comprehension of video content viewed by self-help group (SHG) members. The SHG members' knowledge of the nutrition messages promoted in the videos was high for messages related to IYCF practices. The acceptance and utility of the information were also good [23]. Acceptability is measured using practices and behavior change. Multiple approaches to BCC implemented in North Ethiopia also showed that there was improvement in the practices of mothers regarding empowering new generations to improve nutrition and economic opportunities [24].

The acceptability in this study considered several indicators: i) relevance (i.e., whether the video intervention answers the needs of the mothers and community) [25, 26], ii) how easy it was to implement the intervention/practicability [22, 27, 28]; iii) was it culturally appropriate [25–28], and vi) was it easy to understand [25]. The results of our study are in line with previous studies [22, 28] that assessed acceptability from different perspectives. The messages delivered in the video-based health education of the current trial were reportedly relevant, understandable, culturally appropriate, and easy to implement. The educational package delivered in a video form was locally prepared using multiple approaches, such as testimony, comedy, and dramas and/or roleplays, in the form of questions and answers, group discussions, and deductive approaches. Culturally respectful interventions are more likely to be accepted [22, 27, 28]. A study conducted in Zambia identified that acceptability is increased, and behavior change is enhanced when video-based health education interventions are implemented in the same setting and use similar cultural characteristics [26].

## Feasibility of video-based health education interventions

The feasibility of this type of BCC intervention includes factors ranging from ease (including administrative) of use or implementation [18, 26, 29], technical and logistical capacity (e.g., electric supply) [18, 26], and time dedicated to using the intervention [18, 29]. **Fig 1** provides a

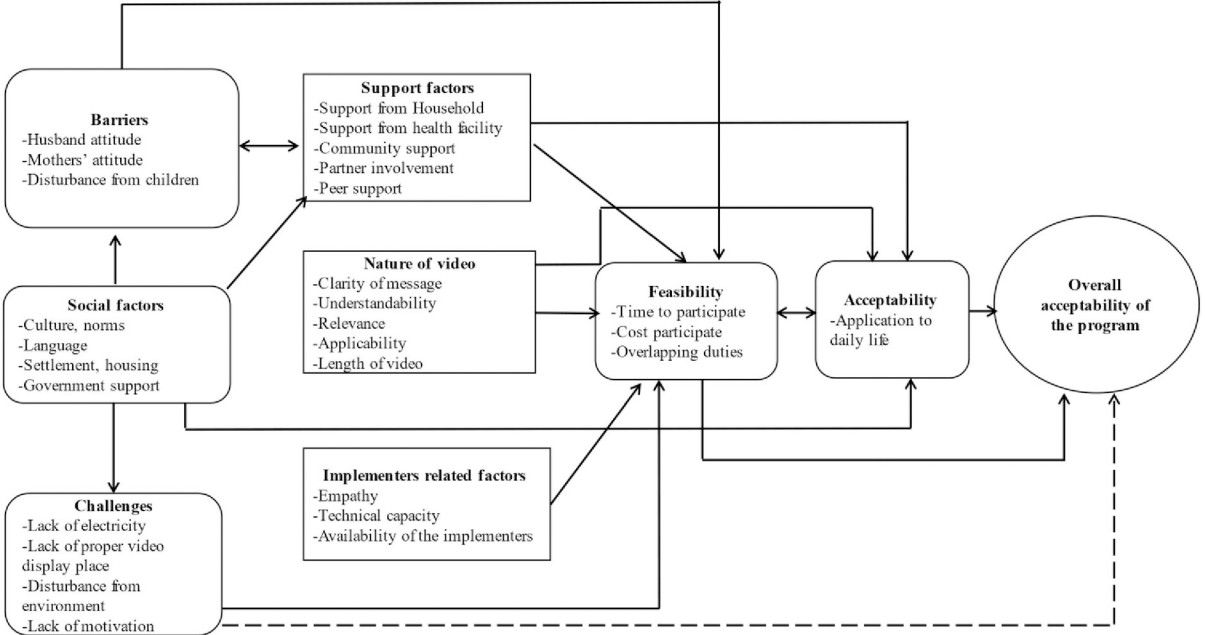

**Fig 1. Thematic map for the acceptability and feasibility of video-based health education for maternal and infant health in the Dirashe District, South Ethiopia.**

summary of facilitators and challenges of acceptability and feasibility of video-based health education for maternal and infant health in the Dirashe District, South Ethiopia.

Fig 1 showing the thematic map for the acceptability and feasibility of video-based health education for maternal and infant health in the Dirashe District, South Ethiopia.

The structured assessment of feasibility (SAFE) tool, which was among the recommended measures of feasibility [30] proposed 16 items for the measurement of feasibility that also affect acceptability, such as culture, additional material resources, staff training, and flexibility. A study in Afghanistan identified that feasibility is also affected by factors that affect acceptability, such as culture, settlement/geography, hard-to-reach areas (access), shortage of time, user friendliness (reduced cost of training), technology (effectiveness), capacity building and support (cost saving), and program design and trainability [18].

Feasibility considers the cost required, the time needed, and the support provided. The videos were projected at the home of the participants, which involved their surroundings, including children, cattle, and the neighborhood, as sources of disturbances. Previous studies on feasibility reported similar findings [30, 31]. Furthermore, our study showed that time consumption and flexibility (or lack thereof) due to overlapping duties were among the constraints preventing participants from attending the video sessions. Matching the prioritized goals in terms of relevance and application to the interests of the participants were facilitators.

## Factors affecting acceptability

Acceptability is also affected by environmental context, household support, and attitude. This was a highly emergent point in our study, and it is in agreement with studies on the acceptability of digital health interventions [15, 22].

In this study, attitude was identified as one of the limiting factors in the acceptability of the video intervention. This is in line with previous findings [15, 27]. Partner attitudes affected the participants' level of commitment, and their involvement in the video intervention was also reported as a key factor in improving the intervention. A similar finding was reported in Uganda, where video-based health education interventions that aimed to enhance perinatal care and uptake of health services were more accepted when the partner was involved and had a supportive attitude [28]. Community and family support are important facilitators of acceptability. A scoping review of acceptability showed that increased support from family and community strengthens the acceptability of interventions [15, 27]. Partner support and their involvement in service uptake after video-based health education interventions were also improved and were measures of acceptability. The other way to measure acceptability is by practicability and taking responsibility [26]. Similarly, the integration of the video project within existing community health services has also been recommended to improve implementation [15, 29, 31].

## Assessment of acceptability using adherence

Acceptability can be assessed using proxies, such as adherence to the intervention. In our study, adherence was measured as attendance at the video projection sessions and willingness to implement the messages provided. A study in Ghana identified acceptability as commonly measured using the level of participant adherence [32]. The other way to measure acceptability is by assessing the impact on behavior, such as confidence to use and implement the intervention among the service providers. Confidence has an effect at work, which can contribute to the effectiveness and attainment of objectives [29]. In addition, knowledge and awareness after the intervention were assessed as part of acceptability and its effects on practicing the behavior

change learned [26]. Behavior changes and acceptance further help service providers improve efficiency and effectiveness in their work [29].

Our study has several strengths, including the timing of the interviews, the diverse profile and involvement of the interviewees and participants, and the innovative aspect of the intervention itself. The interviews were conducted shortly after the end of the intervention and during the data collection period. All the participants in this study contributed as implementers, facilitators, data collectors, or as the target population. However, we recognize the limitations of this study, which include the lack of external validity and the low possibility of generalizing the findings to other interventions and communities in different settings outside of the Ethiopian region. In addition, due to the security reasons, the researchers were not able to meet the participants after analysis was completed, and therefore feedback was not collected.

## Conclusions and implications

The acceptability and feasibility of the video-based health education intervention were good based on the conventional and SAFE guidelines for the measurement of feasibility. This video-based health education can be seen as an alternative way of behavior change communication targeting mothers and their infants. Upscaled successfully, this intervention can overcome the barriers of logistics, as well as community and husband support. Some of the messages should also be tested for feasibility and acceptability to tailor them to local contexts.

## Supporting information

**S1 Checklist. Reporting qualitative studies.**
(DOCX)

**S1 File. Protocol for clinical trial registry.**
(HTML)

**S1 Text. Focus group checklist.**
(DOCX)

**S2 Text. Data collection tools.**
(DOCX)

## Acknowledgments

We are very grateful to the participants in the Dirashe District. Our appreciation goes to the health officials and community health workers for facilitating the implementation of the study. We would like to thank our data collectors, especially Katanso Karso, Godana Kusse, and the health extension workers in the study kebeles, for their support. Finally, we would like to thank our qualitative study participants for their time.

## Author Contributions

**Conceptualization:** Wanzahun Godana Boynito, Stefaan De Henauw, Souheila Abbeddou.

**Data curation:** Wanzahun Godana Boynito, Godana Yaya Tessema, Kidus Temesgen, Souheila Abbeddou.

**Formal analysis:** Wanzahun Godana Boynito, Godana Yaya Tessema.

**Funding acquisition:** Wanzahun Godana Boynito, Stefaan De Henauw, Souheila Abbeddou.

**Investigation:** Wanzahun Godana Boynito, Godana Yaya Tessema, Souheila Abbeddou.

**Methodology:** Wanzahun Godana Boynito, Stefaan De Henauw, Souheila Abbeddou.

**Project administration:** Wanzahun Godana Boynito, Godana Yaya Tessema, Kidus Temesgen, Souheila Abbeddou.

**Resources:** Wanzahun Godana Boynito, Godana Yaya Tessema, Kidus Temesgen, Stefaan De Henauw, Souheila Abbeddou.

**Software:** Wanzahun Godana Boynito, Stefaan De Henauw.

**Supervision:** Wanzahun Godana Boynito, Godana Yaya Tessema, Kidus Temesgen, Stefaan De Henauw, Souheila Abbeddou.

**Validation:** Wanzahun Godana Boynito, Stefaan De Henauw, Souheila Abbeddou.

**Visualization:** Wanzahun Godana Boynito, Souheila Abbeddou.

**Writing – original draft:** Wanzahun Godana Boynito, Godana Yaya Tessema, Kidus Temesgen.

**Writing – review & editing:** Wanzahun Godana Boynito, Godana Yaya Tessema, Kidus Temesgen, Stefaan De Henauw, Souheila Abbeddou.

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
