## [Decision Letter · Decision Letter 0]

8 Sep 2022

PGPH-D-22-01021

Acceptability and feasibility of video-based health education for maternal and infant health in Dirashe district, South Ethiopia: A qualitative study

Dear Dr. Godana,

Thank you for submitting your manuscript to PLOS Global Public Health. After careful consideration, we feel that it has merit but does not fully meet PLOS Global Public Health’s publication criteria as it currently stands. Therefore, we invite you to submit a revised version of the manuscript that addresses the points raised during the review process.

We look forward to receiving your revised manuscript.

Kind regards,

Md Anwarul Azim Majumder, PhD

Academic Editor

Journal Requirements:

2. Please provide separate figure files in .tif or .eps format only and remove any figures embedded in your manuscript file. Please also ensure that all files are under our size limit of 10MB.

3. We noticed that you used "unpublished" in the manuscript. We do not allow these references, as the PLOS data access policy requires that all data be either published with the manuscript or made available in a publicly accessible database. Please amend the supplementary material to include the referenced data or remove the references.

4. We have noticed that you have uploaded Supporting Information files, but you have not included a list of legends. Please add a full list of legends for your Supporting Information files after the references list. 

5. In the online submission form, you indicated that your data will be submitted to a repository upon acceptance.  We strongly recommend all authors deposit their data before acceptance, as the process can be lengthy and hold up publication timelines. Please note that, though access restrictions are acceptable now, your entire data will need to be made freely accessible if your manuscript is accepted for publication. This policy applies to all data except where public deposition would breach compliance with the protocol approved by your research ethics board. If you are unable to adhere to our open data policy, please kindly revise your statement to explain your reasoning and we will seek the editor's input on an exemption. Please be assured that, once you have provided your new statement, the assessment of your exemption will not hold up the peer review process.

Additional Editor Comments (if provided):

Please address the reviewers' comments and improve the paper.

Reviewers' comments:

Reviewer's Responses to Questions

**Comments to the Author**

1. Does this manuscript meet PLOS Global Public Health’s publication criteria? Is the manuscript technically sound, and do the data support the conclusions? The manuscript must describe methodologically and ethically rigorous research with conclusions that are appropriately drawn based on the data presented.

Reviewer #1: Yes

Reviewer #2: Yes

2. Has the statistical analysis been performed appropriately and rigorously?

Reviewer #1: Yes

Reviewer #2: N/A

3. Have the authors made all data underlying the findings in their manuscript fully available (please refer to the Data Availability Statement at the start of the manuscript PDF file)?

Reviewer #1: Yes

Reviewer #2: No

4. Is the manuscript presented in an intelligible fashion and written in standard English?

Reviewer #1: Yes

Reviewer #2: Yes

5. Review Comments to the Author

Reviewer #1: Congratulations to this very well conducted study. I read the manuscript with interest and can highlight that it is well-structured. The methods have been described in detail and are adequate.

However, I do have one major concern which relates to the results section: This section is far too long, mainly because it consists of very many direct quotes. I suggest to delete some of these quotes and describe the results more in a summary. This is a substantial concern which the authors need to address.

Reviewer #2: The article presents acceptability and feasibility of video counseling package for antenatal care, post-pregnancy care for mother and infant in Ethiopia. It appears that this paper presents some part of a trial.

The article needs form refinement and appropriate presentation.

1. Please give some detailed information about the main study/trial and linkage of the qualitative study in the overall study implementation and usage.

2. The rationale for using video based counseling in Ethiopian context should be mentioned in the introduction.

3. The specific comments on the manuscript are marked in the document attached.

6. PLOS authors have the option to publish the peer review history of their article (what does this mean?). If published, this will include your full peer review and any attached files.

**Do you want your identity to be public for this peer review?** For information about this choice, including consent withdrawal, please see our Privacy Policy.

Reviewer #1: **Yes: **Florian Fischer

Reviewer #2: **Yes: **Manoja Kumar Das

---

## [Decision Letter · Decision Letter 1]

13 Feb 2023

PGPH-D-22-01021R1

Acceptability and feasibility of video-based health education for maternal and infant health in Dirashe district, South Ethiopia: A qualitative study

Dear Dr. Godana,

Thank you for submitting your manuscript to PLOS Global Public Health. After careful consideration, we feel that it has merit but does not fully meet PLOS Global Public Health’s publication criteria as it currently stands. Therefore, we invite you to submit a revised version of the manuscript that addresses the points raised during the review process.

The manuscript has been evaluated by ten reviewers, and their comments are available below. Please note that upon resubmission of the revised manuscript, we invited the two previous reviewers but only one of them provided a review. We were therefore required to obtain an additional review from another external reviewer. Unexpectedly, we received eight additional reviews after reaching out to the scientific community.

The reviewers have raised concerns regarding the reporting, methodology and language of this study. Please address all reviewers’ comments where possible. Specifically, we would like to emphasize that several reviewers raised concerns regarding the length of the results section and potentially identifying information about the study participants. 

Could you please revise the manuscript to carefully address the concerns raised?

We look forward to receiving your revised manuscript.

Kind regards,

Johannes Stortz, PhD

Staff Editor

Journal Requirements:

Additional Editor Comments (if provided):

Reviewers' comments:

Reviewer's Responses to Questions

**Comments to the Author**

1. If the authors have adequately addressed your comments raised in a previous round of review and you feel that this manuscript is now acceptable for publication, you may indicate that here to bypass the “Comments to the Author” section, enter your conflict of interest statement in the “Confidential to Editor” section, and submit your "Accept" recommendation.

Reviewer #1: (No Response)

Reviewer #3: (No Response)

Reviewer #4: (No Response)

Reviewer #5: (No Response)

Reviewer #6: (No Response)

Reviewer #7: (No Response)

Reviewer #8: (No Response)

Reviewer #9: (No Response)

Reviewer #10: (No Response)

2. Does this manuscript meet PLOS Global Public Health’s publication criteria? Is the manuscript technically sound, and do the data support the conclusions? The manuscript must describe methodologically and ethically rigorous research with conclusions that are appropriately drawn based on the data presented.

Reviewer #1: Partly

Reviewer #3: Partly

Reviewer #4: Yes

Reviewer #5: Yes

Reviewer #6: Yes

Reviewer #7: Partly

Reviewer #8: Yes

Reviewer #9: Yes

Reviewer #10: Yes

3. Has the statistical analysis been performed appropriately and rigorously?

Reviewer #1: N/A

Reviewer #3: N/A

Reviewer #4: Yes

Reviewer #5: N/A

Reviewer #6: N/A

Reviewer #7: N/A

Reviewer #8: Yes

Reviewer #9: N/A

Reviewer #10: N/A

4. Have the authors made all data underlying the findings in their manuscript fully available (please refer to the Data Availability Statement at the start of the manuscript PDF file)?

Reviewer #1: Yes

Reviewer #3: Yes

Reviewer #4: Yes

Reviewer #5: Yes

Reviewer #6: Yes

Reviewer #7: Yes

Reviewer #8: Yes

Reviewer #9: Yes

Reviewer #10: Yes

5. Is the manuscript presented in an intelligible fashion and written in standard English?

Reviewer #1: Yes

Reviewer #3: Yes

Reviewer #4: Yes

Reviewer #5: No

Reviewer #6: Yes

Reviewer #7: Yes

Reviewer #8: Yes

Reviewer #9: Yes

Reviewer #10: Yes

6. Review Comments to the Author

Reviewer #1: The manuscript has improved but the major issue which I have pointed to in the last round of reviews still remains: The results section mainly included direct quotations. I am missing a synthesis of results going beyond only putting together the quotes from qualitative interviews. There is more abstraction and interpretation needed.

Reviewer #3: In the sampling, authors talk about mothers with good adherence and poor adherence, how was that defined and based on what information

In data collection tools...checklists were used....what checklists and for what purpose

In result theme 1, the acceptability....were there any issues/challenges on acceptability

while this was a qualitative study, languages like determinants should be avoided when possible

The objective was to study "assess feasibility and acceptability" but some points discussed are not in line with that

The topic "quantitative measure...." in the discussion is not relevant since the findings are not clearly discussed, more over this is a qualitative study, discussing about quantitative measures is not good scientific practice.

Reviewer #4: Thank you for the opportunity to review this qualitative study on the acceptability and feasibility of video-based health education for maternal and infant health in South Ethiopia. It was an interesting topic to read about and was well structured.

1. In as much as a summary of the main study implementation has been added to the manuscript, it will benefit from a brief summary of how the videos were shown during the study. From the results section, several scenarios are alluded to: videos shown individually and in groups, at home (or other venues in the community) and in the clinic. It will therefore be easier to contextualize the quotes if there is a summary of how the videos were shown during the course of the study or at the very least, how they were intended to have been shown in the Study setting sub-section.

2. It may be beneficial to briefly explain how ‘adherence’ was defined to determine who qualified as having good or low adherence in the operational definitions section.

3. Line 276 “Mothers unanimously liked the videos, as they are linked to their health.” However, the quote given was about what the women’s least favourite video was. You may want to consider using a quote that will better reflect the sub-theme of what was unanimously liked (or change the sub-theme to what the mothers did not like about the videos and present the quote(s) to buttress this sub-them).

4. In the discussion section, you gave a hypothesis which is the foundation of your discussion (lines 528-585). Please add the citation(s) for this hypothesis.

5. A couple of the themes in the Figure 1 were not really explored in the results or the discussion: political environment and empathy of implementers. It will be beneficial to explore these themes further in the manuscript if they were indeed themes that emerged or justify their presence in the figure.

6. A minor point: A few quotes had full participant names attached to them, for example, on line 381. Better to give only the 1st name as is best practice and has been done throughout the paper. On the same line, participant was also identified as “FGD” without a description of what their role was (FGD discussant?).

Reviewer #5: This qualitative study demonstrated acceptability and feasibility of video-based health education intervention for maternal and infant health through FGDs and KIIs among different stakeholders in a district in South Ethiopia.

1) Although the authors tried to address the issues highlighted in the first review, the major concerns related to length of result section remains yet. I would like to suggest author to interpret the results more precisely, particularly for the long quotes.

2) Author should be specific about their intervention. They mentioned intervention in different terms like “community-based video intervention”, “video-based intervention”, or “video-based health education”. It should be consistent in whole manuscript. It seems to be a video-based health education intervention, suggested to replace by it.

3) Please use full-form when you mentioned time (e.g., 1 hour 25 minutes to 1 hour 45 minutes in line 167, 35 minutes to 55 minutes in line 168).

4) There are several typo-errors (e.g., 37 KIIs in line 23, 10% in line 175, and so on), please check the whole manuscript.

5) Please check reference styles, few references seem to be inconsistent. Author used both full-form and abbreviation of journal name.

Reviewer #6: 1. On L28 and 32the word “good” needs to be clarified.

2. Please remove L151-153- you don’t need to quantify qualitative research.

3. The result section still needs trimming and ramifications.

4. Please interpret the data and put a quote that supports the point you wanted to make. Please select a single quote from one study group that can clearly elaborate on the idea you want to raise.

5. Walayte (FGD discussant)- Please correct the name of the place.

6. Please use a similar labelling format for the quotes. E. g (HDA), (FGD discussant), (FGD) Mention either the study type, the respondent type or both uniformly.

7. Please remove personal identifiers – e.g “Kolla Mashile” if these are names of a district, please mention the district names in the methods section.

8. Please reduce the length of some lengthy quotes. Go for a maximum of 6 lines.

Reviewer #7: The manuscript presents a qualitative study conducted at the end of a Cluster Randomized Community Trial. It is an appropriate complement to the main study as it endeavors to evaluate the intervention implemented. On the whole, study focus, logical flow and cohesion of the paper needs improvement. General areas of improvement are as follows:

1. Study focus

To enhance understanding and appreciation of the manuscript, the focus/objectives of this qualitative study need to be made very clear in the introduction and run through the other sections of the write up (methodology, results, discussion and conclusion) to ensure logical flow of the manuscript. The variables of acceptability and feasibility as well as the methods and tools of their assessment also need to be explicitly stated. Differing objectives and variables are alluded to in different sections of the manuscript making it challenging to discern the core study focus.

2. Introduction

This should be shortened to succinctly make a case for the qualitative study in relation to

• Why digital innovative methods to facilitate nutrition education and counselling need to be in place

• Importance of looking at the feasibility and acceptability of the video-based health education methods and their linkage to promoting behavior change at scale

• The limitation of information provided in ANC.

It is unclear what beliefs, attitudes, needs and situations were being investigated. Furthermore, aspects of identifying opportunities and challenges are also referred to but results related to this are not reflected in the abstract nor results section of the paper. Study focus needs to be appropriately synchronized in the title and main body of the paper.

3. Materials and methods

Expound on specific procedures, tools and variables of interest for the qualitative study.

4. Results

• Provide evidence-based results from the FGDs and KIs that are well linked to the objectives and variables of interest

• Need to standardize labelling of quotes by location, data collection method and respondent type.

• Ensure privacy and confidentiality of respondent is safeguarded

5. Discussion should

• be closely linked to the result

• logically flow as per the objectives/results presented without intermingling different issues

• show linkage or difference with similar studies done in the same or different context.

6. Conclusion and implications

• Conclusions should be drawn from the study results

• Implication needs to allude to if acceptability and feasibility was good, what does that mean for maternal and infant health

7. Check grammar and sentence construction ensuring logical connection between the sentences and paragraphs

Specific comments are provided in the attached document.

Reviewer #8: Based on the findings in the research, I agree with the authors claim that the acceptability and feasibility of video-based health education for maternal and infants in Dirashe district is good. I think that the results are strongly supported with the data provided.

I only have minor edits (in tract changes) and comments. I would prefer to reduce the quotations in each theme. About two or maximum of three should be enough to support a theme. For example, in line 288- 'Feasibility of video-based health education among pregnant and lactating women...': there are five quotations which could be reduced to two or a maximum of three of different participants.

My overall impression is that the paper is strong and i recommend for it to be published with these minor edits/comments.

Reviewer #9: This significant research project uses a phenomenological qualitative study design to examine the feasibility and acceptability of video-based behavioral change communication in the local setting with the goal of enhancing both the health of the mother and the newborn Overall, the manuscript is highly fascinating, straightforward, and condensed. Address the aforementioned suggestions to make the manuscript more valuable.

Reviewer #10: (No Response)

7. PLOS authors have the option to publish the peer review history of their article (what does this mean?). If published, this will include your full peer review and any attached files.

**Do you want your identity to be public for this peer review?** For information about this choice, including consent withdrawal, please see our Privacy Policy.

Reviewer #1: **Yes: **Florian Fischer

Reviewer #3: No

Reviewer #4: **Yes: **Grace Christopher Mambula

Reviewer #5: **Yes: **Md. Obaidur Rahman

Reviewer #6: **Yes: **Anene Tesfa Berhanu

Reviewer #7: **Yes: **Gakenia Wamuyu Maina

Reviewer #8: **Yes: **Yusupha Dibba

Reviewer #9: **Yes: **Trhas Tadesse Berhe

Reviewer #10: No

---

## [Decision Letter · Decision Letter 2]

9 May 2023

Acceptability and feasibility of video-based health education for maternal and infant health in Dirashe district, South Ethiopia: A qualitative study

PGPH-D-22-01021R2

Dear Godana,

We are pleased to inform you that your manuscript 'Acceptability and feasibility of video-based health education for maternal and infant health in Dirashe district, South Ethiopia: A qualitative study' has been provisionally accepted for publication in PLOS Global Public Health.

Best regards,

Julia Robinson

Executive Editor

Reviewer Comments (if any, and for reference):

Reviewer's Responses to Questions

**Comments to the Author**

1. If the authors have adequately addressed your comments raised in a previous round of review and you feel that this manuscript is now acceptable for publication, you may indicate that here to bypass the “Comments to the Author” section, enter your conflict of interest statement in the “Confidential to Editor” section, and submit your "Accept" recommendation.

Reviewer #1: (No Response)

Reviewer #4: All comments have been addressed

Reviewer #5: All comments have been addressed

Reviewer #7: All comments have been addressed

2. Does this manuscript meet PLOS Global Public Health’s publication criteria? Is the manuscript technically sound, and do the data support the conclusions? The manuscript must describe methodologically and ethically rigorous research with conclusions that are appropriately drawn based on the data presented.

Reviewer #1: (No Response)

Reviewer #4: (No Response)

Reviewer #5: Yes

Reviewer #7: Yes

3. Has the statistical analysis been performed appropriately and rigorously?

Reviewer #1: (No Response)

Reviewer #4: (No Response)

Reviewer #5: N/A

Reviewer #7: N/A

4. Have the authors made all data underlying the findings in their manuscript fully available (please refer to the Data Availability Statement at the start of the manuscript PDF file)?

Reviewer #1: (No Response)

Reviewer #4: (No Response)

Reviewer #5: Yes

Reviewer #7: Yes

5. Is the manuscript presented in an intelligible fashion and written in standard English?

Reviewer #1: (No Response)

Reviewer #4: (No Response)

Reviewer #5: Yes

Reviewer #7: Yes

6. Review Comments to the Author

Reviewer #1: The result section has improved a lot.

Reviewer #4: (No Response)

Reviewer #5: Author made significant revision and tried to address all reviewers' comments. I don't have further comment.

Reviewer #7: The authors have adequately addressed the previous comments provided

7. PLOS authors have the option to publish the peer review history of their article (what does this mean?). If published, this will include your full peer review and any attached files.

**Do you want your identity to be public for this peer review?** For information about this choice, including consent withdrawal, please see our Privacy Policy.

Reviewer #1: **Yes: **Florian Fischer

Reviewer #4: **Yes: **Grace Christopher Mambula

Reviewer #5: **Yes: **Dr. Md. Obaidur Rahman

Reviewer #7: **Yes: **Gakenia Wamuyu Maina
